# Morphological Characteristics and Situational Precision of U15 and U16 Elite Male Players from Al-Ahli Handball Club (Bahrein)

**DOI:** 10.3390/sports10070108

**Published:** 2022-07-06

**Authors:** Boris Banjevic, Boris Zarkovic, Borko Katanic, Blazo Jabucanin, Stevo Popovic, Bojan Masanovic

**Affiliations:** 1Faculty for Sport and Physical Education, University of Montenegro, 81400 Niksic, Montenegro; boris.banjevic@gmail.com (B.B.); bojanma@ucg.ac.me (B.M.); 2Faculty of Sport and Physical Education, University of Novi Sad, 21000 Novi Sad, Serbia; zarkovicboris@gmail.com; 3Faculty of Sports and Physical Education, University of Nis, 18000 Nis, Serbia; borkokatanic@gmail.com; 4Western Balkan Sport Innovation Lab, 81000 Podgorica, Montenegro; blazojabucanin@yahoo.com

**Keywords:** handball, specific precision, anthropometry, longitudinal dimensionality

## Abstract

The aim of this cross-sectional study was to determine the differences in the morphological characteristics and situational precision among younger and older groups of handball players. The sample of participants consisted of 30 handball players, members of the younger category of the Al-Ahli handball club (Bahrein), divided into two groups: older (U16, *n* = 18) and younger (U15, *n* = 12). To evaluate their morphological characteristics, eight variables were measured, while two standardized tests were used to evaluate their situational precision. The results indicate that a statistically significant difference between the groups was noticeable for nine variables in total, seven in morphology (body height, *p* = 0.010; body mass index, *p* = 0.049; arm length, *p* = 0.009; upper arm length, *p* = 0.016; lower arm length, *p* = 0.040; the planimetric parameter of the hand, *p* = 0.005; hand length *p* = 0.004) and two in situational precision (the standing shot, *p* = 0.003; the jump shot, *p* = 0.17), and that the achieved difference ranges from a medium to a large effect. For only one variable (body mass, *p* = 0.734), significant difference was not determined between the groups. It was also determined (by Cohen’s criterion) that handball players with higher longitudinal dimensionality achieve better results for specific precision. Therefore, when selecting young handball players, the aforementioned dimensions should be taken into consideration as predictors of success.

## 1. Introduction

Current research indicates that the morphological characteristics of athletes are significantly associated with success in sport [1,2,3,4,5]. In relation to the demands of a certain sport, every athlete should possess optimal morphological characteristics [4,6,7,8], and therefore Milanovic et al. [9] cite that their determination represents one of the three most frequently tested dimensions of athletes. Insight into the morphological characteristics of athletes has practical significance when adapting the training process to the individual characteristics of an athlete, as well as during the evaluation of their ultimate abilities [10]. When it comes to young athletes, Masanovic and Vukasevic [5] point out that they too must have optimal morphological characteristics and motor skills so as to meet the requirements of a certain sport. It should be added that morphological characteristics of young athletes are susceptible to change and can to a significant extent be improved [11].

Handball is a demanding contact sport which in its structure contains elements of movement such as running, jumping, sprinting, swinging, hitting, blocking and pushing an opponent [12]. Consequently, the performance of handball players is affected by morphological characteristics, technical and tactical skills, and physical abilities which develop with age [13]. Among young handball players of various ages, the determination of morphological measurements represents an important part of the effective modelling of the training process and selection [14]. 

Current research carried out on handball players of various age categories and levels of competition indicates that handball requires higher values of body height and body mass [15]. Even though authors have dealt with the morphological differences between young handball players and other athletes such as soccer players, basketball players, and volleyball players [8], there are still not enough studies which have analyzed differences in the morphological features among young handball players in relation to age categories. 

Moreover, one of the most important skills for success in handball is the ability to throw [16] and for that reason numerous studies have focused on the throwing technique [16,17,18,19]. It was determined that success in throwing is affected by precision [17,18,19] and speed of the ball [16,19]. Precision and the speed of the throw are increasingly gaining in importance in the end results of a game and represent two basic factors, both of which are important for the effectiveness of the throw in handball [20]. Despite this, certain authors point out that precision is more important, and that if athletes focus on precision, speed decreases [21]. Still, insufficient data are available regarding the precision of handball players, especially handball players in younger categories. 

Therefore, this study aim to determine the differences in the morphological characteristics and situational precision between older and younger male groups of handball players.

## 2. Methods

### 2.1. The Sample of Participants

The sample of participants consisted of 30 handball players, all members of younger categories of the Al-Ahli club (Bahrein), divided into two groups: an older group of boys born in 2002 (*n* = 18, 15.60 ± 0.30 yrs.) and a younger group of boys born in 2003 (*n* = 12, 14.69 ± 0.32 yrs.). The sample included participants who were psychologically and physically healthy, who had no pronounced aberrations concerning the locomotor apparatus. In addition, the inclusion criteria for the participants were that they have actively been taking part in handball for at least a period of 3 years, that they train at least 4 times a week, and that are competing in the appropriate age category. All of the participants, as well as their parents, were well acquainted with the procedure and purpose of the testing, and the parents gave their consent on the condition that they could withdraw their children from the study at any point (which none of them did). All procedures performed in studies involving human participants were in accordance with the 1964 Helsinki Declaration and its later amendments or comparable ethical standards.

Skinfold thicknesses (mm) were measured at six sites: triceps skinfold, forearm skinfold, thigh skinfold, calf skinfold, chest skinfold, and abdominal skinfold thickness (using a skinfold caliper).

### 2.2. The Sample of Measuring Instruments

Standardized anthropometric instruments were used to measure the morphological characteristics. The measurements were taken according to a predetermined International Biological Program (IBP) [22]. To evaluate the morphological characteristics, eight variables were measured: body height, body mass, body mass index, arm length, upper arm length, lower arm length, the planimetric parameter of the hand, and hand length [23]. Height was measured to the nearest 0.1 cm using a fixed stadiometer (Seca, Leicester, UK) and weight was measured to the nearest 0.1 kg with an electronic weighing machine (HD-351, Tanita, Illinois, USA). The body mass index was calculated using the standard formula: BMI = body mass (kg) ÷ body height2 (meters). Length measures were measured to the nearest 0.1 cm, and taken with the Martin anthropometer (GPM, Bachenbülach, Switzerland). At the same time, to evaluate the situational precision of the handball players, two standardized situational-motor tasks of a composite type were used, selected on the basis of existing studies: precision of the throw from a standing position on both legs from 7 m, and precision of the throw from a jump at 9 m [24,25,26,27]. The protocol is the same for both tests, and the target is a unilateral triangle whose sides are 50 cm in length, while the hypotenuse is an elastic band 5 cm wide (Figure 1). Each test consists of seven attempts of four shots each into every corner of the goal (the participant makes the shots in the following sequence: the upper left-hand corner, the upper right-hand corner, the lower left-hand corner, the lower right-hand corner). Every shot that landed within the triangle carried two points, the ones that landed on the frame of the triangle one point, and a miss carried zero points. The result represents the sum of all the points from all seven attempts. 

### 2.3. Statistical Analysis

The data analysis was carried out using the IBM SPSS Statistics 26 software (Chicago, IL, USA). The means and standard deviation were calculated for each variable. To evaluate the normality of the distribution, the Shapiro-Wilk’s test was used. Differences in the morphological characteristics and situational precision between the groups were determined using a univariate analysis of variance (ANOVA), while the level of significance was set at *p* < 0.05. What was then calculated was Eta squared (η^2^), as an indicator of effect size. Cohen’s classification indicates the size of the effect whereby 0.01 is a small effect, 0.06 is a medium effect and 0.14 is a large effect [28].

## 3. Results

The normal distribution of the data was confirmed by the Shapiro-Wilk test (Table 1). One variable indicated a deviation from normal distribution (Lower arm length, *p* = 0.021), while the remaining variables have normal distribution (*p* > 0.05) and, along with a visual overview of the histogram and Q-Q plot, it was determined that the data were normal, and that a parametric technique could be used.

When we view the average values (Mean) between the younger and older group of handball players (Table 2), it can be noted that the older group achieved greater values for all the variables which pertain to morphological measurements and situational precision, except for the body mass index for which higher values were noted for the younger group. 

The ANOVA determined that only the values of body mass did not show a significant difference (*p* > 0.05) between the older and younger group of handball players, while a significant difference between the groups was noted for the remaining nine variables (*p* = 0.003–0.049), which also goes for the seven morphological variables (*p* = 0.004–0.49), and two variables of situational precision, the standing shot (*p* = 0.003) and the jump shot (*p* = 0.17).

Based on Eta squared and Cohen’s criterion [28], we can note that the actual difference between the means for the given nine variables has a medium to large effect, of which the variable for the body mass index has medium effect size (η^2^ = 0.131), while all the other variables have a large effect size (η^2^ = 0.142–0.275).

## 4. Discussion

This study showed that there are differences between the morphological characteristics of the two studied groups of young handball players, members of the Al-Ahli handball club, and also between their results for situational precision. Statistically significant differences with greater values for the older boys were noted for all the variables which refer to the longitudinal dimensionality of the skeleton, which is congruent with the results of Vuleta et al. [14], who also noted higher values for the variables of longitudinal dimensionality for older boys. Considering the difference in chronological age and that the participants are in a period of turbulent growth and development, which is still enabling the growth of various dimensions of the body [29], it could be said that these differences were expected. Contrary to this, a statistically significant difference for the BMI variable, with higher values for the young group of boys, was not expected. Even though the intensity and scope of training were identical for both groups of handball players, greater values of BMI were noted for the group of younger players, which can be ascribed to an inappropriate diet, biological age, and genetic conditioning. Based on the descriptive data which refer to body height and body mass, we can see that the data for young handball players from Al-Ahlija do not deviate much from the results of similar studies which were carried out locally [30,31]. If we were to compare the morphological characteristics of Bahraini young handball players to those of their peers from Croatia and Serbia [14,32], who have been at the top of European sports for years, we could say that, despite the decreased population from which young handball players are extracted [33], Bahraini coaches do carry out proper talent identification. 

When we look at the results of tests which measured the specific precision of young handball players (the standing shot, and the jump shot) we can see that there were statistically significant differences on both tests with better results scored by the older group of boys. Šibila and Pori [34] point out that handball at an elite level requires certain morphological characteristics which are reflected in the longitudinal dimensionality of the skeleton, while Taborsky [35] states that greater dimensionality of the hand leads to better control of the ball. According to their claims and the analysis of the results of anthropometric measurements which refer to the size of the hand and the results on specific tests of precision, our results suggested that handball players with greater longitudinal dimensionality of the skeleton scored better results. This observation is also congruent with previous research which dealt with the same topic [26,27]. The mentioned differences can also be conditioned by preforming a throw in situational conditions with increased speed. During fast movements, the possibility of visual and kinesthetic monitoring is significantly reduced, which ultimately reflects on the technique and precision of performance [36]. Given that older handball players have many years of practice and greater handball experience, and therefore significantly better and more consistent sports technique, it was to be expected that they would achieve better results in situational precision.

In order to compare the precision of young Bahraini handball players, which along with the strength of the throw is the most important component of an effective throw in handball [19], there is a need for such studies to be carried out on larger samples and on handball players who compete in environments where elite handball is played. Such studies could be used to evaluate the quality of the training process itself, that is, the level of technical skills of young Bahraini handball players compared to European ones. In addition, the players should also be classified based on their position on the team, since satisfactory knowledge of the morphological characteristics of players for each position is of key importance for coaches when it comes to the orientation of players in certain positions. Accordingly, when selecting young handball players, the aforementioned dimensions should be taken into consideration as predictors of success.

## Figures and Tables

**Figure 1 sports-10-00108-f001:**
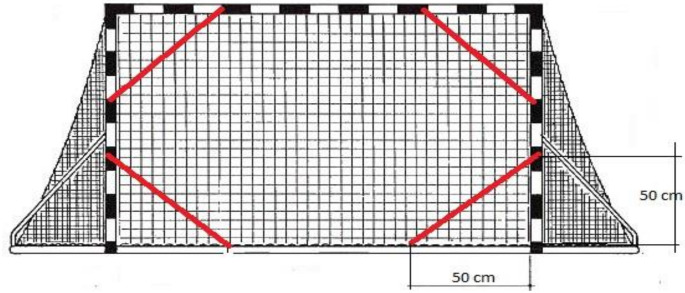
Marked targets on the goal.

**Table 1 sports-10-00108-t001:** The Shapiro-Wilk test of normality.

	Shapiro-Wilk	
	Statistic	Sig.
Body height (cm)	0.961	0.326
Body mass (kg)	0.969	0.512
BMI (score)	0.974	0.650
Arm length (cm)	0.954	0.216
Upper arm length (cm)	0.953	0.203
Lower arm length (cm)	0.916	0.021
Planimetry of the hand(cm)	0.950	0.166
Hand length (cm)	0.948	0.154
The standing shot (score)	0.965	0.420
The jump shot (score)	0.951	0.184

Legend: BMI = body mass index.

**Table 2 sports-10-00108-t002:** Descriptive data and the differences between younger and older groups handball players (ANOVA).

	Older (*n* = 18)	Younger (*n* = 12)	ANOVA		
Variables	Mean ± SD		F	*p*	η^2^
Body height (cm)	174.28 ± 4.57	168.00 ± 7.90	7.628	0.010 *	0.211
Body mass (kg)	56.67 ± 4.98	56.03 ± 5.05	0.118	0.734 ^	0.004
BMI (score)	18.66 ± 1.53	19.88 ± 1.69	4.219	0.049 *	0.131
Arm length (cm)	78.01 ± 1.65	76.06 ± 2.14	7.963	0.009 *	0.221
Upper arm length (cm)	31.31 ± 1.33	29.78 ± 1.92	6.611	0.016 *	0.191
Lower arm length (cm)	29.49 ± 1.00	28.30 ± 2.02	4.637	0.040 *	0.142
Planimetry of the hand (cm)	22.23 ± 1.39	20.87 ± 0.88	9.120	0.005 *	0.246
Hand length (cm)	19.13 ± 0.95	17.88 ± 1.25	9.652	0.004 *	0.256
The standing shot (score)	30.44 ± 3.59	26.17 ± 3.43	10.598	0.003 *	0.275
The jump shot (score)	22.06 ± 5.39	16.58 ± 6.30	6.492	0.017 *	0.188

Legend: Mean = Arithmetic mean; SD = Standard deviation; F = ANOVA test value; *p* = Statistical significance; η^2^ = Eta square; ^ = Non-significant; * = Significant difference.

## Data Availability

Not applicable.

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
