# Peer review of "Morphological Characteristics and Situational Precision of U15 and U16 Elite Male Players from Al-Ahli Handball Club (Bahrein)"

_sports, 2022, doi:10.3390/sports10070108_

Round 1
Reviewer 1 Report
Article ID: sports-1744298
Title: Morphological Characteristics and Situational Precision of U15 and U16 Elite Handball Players from Bahrain
Specific comments
TITLE
Suggestion: Morphological Characteristics and Situational Precision of U15 and U16 Elite Male Players from Al-Ahli Handball Club (Bahrein)
ABSTRACT
P1-Line 15-6: divided into two groups: older (n=18, 15.60±.30 yrs.) and younger (n=12, 15 14.69±.32 yrs.). // divided into two groups: older (U16, n=18) and younger (U15, n=12).
P1-Line 16: Morphological // morphological
INTRODUCTION
P1-Line 31: (Malina et al, 2004; // (Malina et al., 2004;
P1-Line 35: (Milanović et al, 2005) // (Milanović et al., 2005)
P1-Line 38: Masanovic & Vukasevic (2020) // Masanovic and Vukasevic (2020)
P1-L39-40: point out that they too must have optimal morphological characteristics and motor skills so as to meet the requirements of a certain sport. // point out that they too must have optimal morphological characteristics and motor skills so as to meet the requirements of a certain sport.
P1-L41-42: and can to a significant extent be improved // and can to a significant extent be improved
P2-Line 57: Suggestion: One of the most // Moreover, one of the most
P2-Line 69-71: Suggestion: the aim of this study was to determine the morphological characteristics and situational precision of the handball cadets, as well as to study the differences in the given characteristics between older and younger cadet handball players. // this study aim to determine the differences in the morphological characteristics and situational precision between older and younger male cadet handball players.
METHODS
P2-Line 91: length. // hand length.
P2-Line 92: BMI=TM(kg)/TV(m)2. // BMI = body mass (kg) ÷ body height2 (meters)
P2-Line 92: Please provide (also) a brief description of: arm length, upper arm length, lower arm length, the planimetric parameter of the hand, and hand length. Complementarily, present the references of used instruments in data collection (and ICCs).
P3-Line 108: Methods of data processing // Statistical analysis
P3-Line 119: normal distribution n (DUPO, p=.021), // normal distribution (Lower arm length, p=.021), //
P4-Line 124-5: Legend: n= number of subjects; BMI= body mass index; ^= non-significant; *= significant difference 124 between the groups. // Legend: BMI= body mass index.
DISCUSSION
P4-Line 154-58: Based in the previous sentence, authors can comment the impact of growth (and PHV) on BMI changes.
P5-Line 169: Šibila & Pori // Šibila and Pori
P5-Line 174-5: we can conclude … // , our results suggested …
P5-Line 176: This conclusion is also // This observation is also
P5-Line 181-3: “Given that older handball players have significantly better and more 181 consistent sports technique, it was to be expected that they would achieve better results 182 in situational precision.” // Authors need to discuss this sentence in order to years of practice, handball experience, etc..
P5-Line 194: To conclude, in this study // In sum,
P5-Line 195: it was possible to conclude // it seems
Author Response
TITLE
Suggestion: Morphological Characteristics and Situational Precision of U15 and U16 Elite Male Players from Al-Ahli Handball Club (Bahrein)
Done! Thank you!
ABSTRACT
P1-Line 15-6: divided into two groups: older (n=18, 15.60±.30 yrs.) and younger (n=12, 15 14.69±.32 yrs.). // divided into two groups: older (U16, n=18) and younger (U15, n=12).
Done! Thank you!
P1-Line 16: Morphological // morphological
Done! Thank you!
INTRODUCTION
P1-Line 31: (Malina et al, 2004; // (Malina et al., 2004;
Done! Thank you!
P1-Line 35: (Milanović et al, 2005) // (Milanović et al., 2005)
Done! Thank you!
P1-Line 38: Masanovic & Vukasevic (2020) // Masanovic and Vukasevic (2020)
Done! Thank you!
P1-L39-40: point out that they too must have optimal morphological characteristics and motor skills so as to meet the requirements of a certain sport. // point out that they too must have optimal morphological characteristics and motor skills so as to meet the requirements of a certain sport.
Done! Thank you!
P1-L41-42: and can to a significant extent be improved // and can to a significant extent be improved
Done! Thank you!
P2-Line 57: Suggestion: One of the most // Moreover, one of the most
Done! Thank you!
P2-Line 69-71: Suggestion: the aim of this study was to determine the morphological characteristics and situational precision of the handball cadets, as well as to study the differences in the given characteristics between older and younger cadet handball players. // this study aim to determine the differences in the morphological characteristics and situational precision between older and younger male cadet handball players.
Done! Thank you!
METHODS
P2-Line 91: length. // hand length.
Done! Thank you!
P2-Line 92: BMI=TM(kg)/TV(m)2. // BMI = body mass (kg) ÷ body height2 (meters)
Done! Thank you!
P2-Line 92: Please provide (also) a brief description of: arm length, upper arm length, lower arm length, the planimetric parameter of the hand, and hand length. Complementarily, present the references of used instruments in data collection (and ICCs).
We presented the references of used instruments, and also add reference of the standard international protocol according to which we performed anthropometric measurements.
P3-Line 108: Methods of data processing // Statistical analysis
Done! Thank you!
P3-Line 119: normal distribution n (DUPO, p=.021), // normal distribution (Lower arm length, p=.021), //
Done! Thank you!
P4-Line 124-5: Legend: n= number of subjects; BMI= body mass index; ^= non-significant; *= significant difference 124 between the groups. // Legend: BMI= body mass index.
Done! Thank you!
DISCUSSION
P4-Line 154-58: Based in the previous sentence, authors can comment the impact of growth (and PHV) on BMI changes.
Thank you for the advice, but we already commented the impact of growth on morphological and BMI changes.
P5-Line 169: Šibila & Pori // Šibila and Pori
Done! Thank you!
P5-Line 174-5: we can conclude … // , our results suggested …
Done! Thank you!
P5-Line 176: This conclusion is also // This observation is also
Done! Thank you!
P5-Line 181-3: “Given that older handball players have significantly better and more 181 consistent sports technique, it was to be expected that they would achieve better results 182 in situational precision.” // Authors need to discuss this sentence in order to years of practice, handball experience, etc..
Done! Thank you!
P5-Line 194: To conclude, in this study // In sum,
Done! Thank you!
P5-Line 195: it was possible to conclude // it seems
Done! Thank you!

Reviewer 2 Report
Abstract
This section gives the future reader a good idea of what they will find in the article. However, it is useful to include some of the data where you report that there are significant differences between groups.
Introduction
Lines 30-32. If you say they are "Current research" it is not advisable to include only two citations and that one is from 2004. This should be thoroughly reconsidered and many more references, especially meta-analyses, should be included.
Line 33. Same with the quote from (Hatzimanouil & Oxizoglou, 2004).
Lines 33-35. If you indicate that it is a review of authors, in plural, you cannot put only one citation (Milanović et al, 2005) and that it is so old. In general, it is advisable not to include citations older than 5 years. Please check throughout the text.
Lines 36-37. Avoid breaking syllables at the end of a line. This is a spelling mistake and yes we know that the journal template allows it, but this does not mean that we refuse to write correctly. Please check and change throughout the text.
Lines 54-55. There is an excessive citation of Masanovic who is also the author of the paper.
Lines 58-60. When you include this statement "numerous studies have recently focused" ... and the citations are from 2003-2009!!!!... They are not recent. Please change and include really recent citations.
In general the introduction attracts the reader's attention well but it lacks the removal of self-citations and the inclusion of more up to date references.
2. Methods
2.1. The sample of participants
Please include the code of ethical approval of the committee that approved the research and report whether international protocols for research involving human subjects were respected.
2.2. The sample of measuring instruments
In this section you should report exactly what the measuring instruments were, including their brand and level of precision.
2.3. Methods of data processing
This section is well explained and the analyses relevant to this research have been used.
3. Results
In tables 1 and 2 you should include the units of measurement for each of the variables, e.g. Body height (cm); Body mass (kg) ...
Lines 124-125. In the legend of table 1 it does not make sense to include ^= non-significant; *= significant difference 124 between the groups.
Lines 142-143. In the legend, as you did in the previous table, you should change the dashes (-) to the equal sign (=). This is the usual and has been included before and in Non-significant.
4. Discussion
The authors make a genuine effort to compare their results with those of previous research even though they recognise that their sample is relatively small and only from one club.
In lines 95-96 they report that the throws are only made from 9 metres, however, in line 168 they report that there are two accuracy tests (SST7 and SS9M) of which nothing is reported in the methodology or results.
As for the conclusion reached in lines 175-176 "that handball players with greater longitudinal dimensionality of the skeleton scored better results". It would have been necessary to carry out a bivariate correlation analysis in order to verify whether this relationship between anthropometry and accuracy has any statistical basis.
Author Response
Abstract
This section gives the future reader a good idea of what they will find in the article. However, it is useful to include some of the data where you report that there are significant differences between groups.
Done! Thank you!
Introduction
Lines 30-32. If you say they are "Current research" it is not advisable to include only two citations and that one is from 2004. This should be thoroughly reconsidered and many more references, especially meta-analyses, should be included.
Done! Thank you!
Line 33. Same with the quote from (Hatzimanouil & Oxizoglou, 2004).
Done! Thank you!
Lines 33-35. If you indicate that it is a review of authors, in plural, you cannot put only one citation (Milanović et al, 2005) and that it is so old. In general, it is advisable not to include citations older than 5 years. Please check throughout the text.
Done! Thank you!
Lines 36-37. Avoid breaking syllables at the end of a line. This is a spelling mistake and yes we know that the journal template allows it, but this does not mean that we refuse to write correctly. Please check and change throughout the text.
Done! Thank you!
Lines 54-55. There is an excessive citation of Masanovic who is also the author of the paper. Done! We have reduced the number of citations by Masanovic from 5 to 2.
Lines 58-60. When you include this statement "numerous studies have recently focused" ... and the citations are from 2003-2009!!!!... They are not recent. Please change and include really recent citations.
Done! Thank you!
In general the introduction attracts the reader's attention well but it lacks the removal of self-citations and the inclusion of more up to date references.
Done! Thank you!
- Methods
2.1. The sample of participants
Please include the code of ethical approval of the committee that approved the research and report whether international protocols for research involving human subjects were respected.
Done! Thank you!
2.2. The sample of measuring instruments
In this section you should report exactly what the measuring instruments were, including their brand and level of precision.
Done! Thank you!
2.3. Methods of data processing
This section is well explained and the analyses relevant to this research have been used.
Thank you!
- Results
In tables 1 and 2 you should include the units of measurement for each of the variables, e.g. Body height (cm); Body mass (kg) ...
Done! Thank you!
Lines 124-125. In the legend of table 1 it does not make sense to include ^= non-significant; *= significant difference 124 between the groups.
We left this because we think it is important to emphasize what does it mean.
Lines 142-143. In the legend, as you did in the previous table, you should change the dashes (-) to the equal sign (=). This is the usual and has been included before and in Non-significant.
Done! Thank you!
- Discussion
The authors make a genuine effort to compare their results with those of previous research even though they recognise that their sample is relatively small and only from one club.
Thank you!
In lines 95-96 they report that the throws are only made from 9 metres, however, in line 168 they report that there are two accuracy tests (SST7 and SS9M) of which nothing is reported in the methodology or results.
Done! Thank you!
As for the conclusion reached in lines 175-176 "that handball players with greater longitudinal dimensionality of the skeleton scored better results". It would have been necessary to carry out a bivariate correlation analysis in order to verify whether this relationship between anthropometry and accuracy has any statistical basis.
Done! Thank you!

Round 2
Reviewer 2 Report
The results that have been included in the abstract are values that it is impossible for the reader to understand which variables they belong to.
In the Introduction it can be considered that the requested corrections have been made.
In the Methodology it can be considered that the requested corrections have been made.
In the Results it can be considered that the requested corrections have been made.
In the Discussion, the requested corrections can be considered to have been made.
Author Response
Dear reviewer,
thank you very much for you comments. We agree the abstract could be written better, so we revised it and we do believe this version is much better and acceptable for you.
The results that have been included in the abstract are values that it is impossible for the reader to understand which variables they belong to.
Thanks! Have done.
In the Introduction it can be considered that the requested corrections have been made.
Thanks.
In the Methodology it can be considered that the requested corrections have been made.
Thanks.
In the Results it can be considered that the requested corrections have been made.
Thanks.
In the Discussion, the requested corrections can be considered to have been made.
Thanks.